# Neck Circumference as a Screening Tool for Metabolic Syndrome among Lebanese College Students

**DOI:** 10.3390/diseases10020031

**Published:** 2022-06-02

**Authors:** Suzan A. Haidar, Nanne de Vries, Kalliopi-Anna Poulia, Hussein Hassan, Mohammad Rached, Mirey Karavetian

**Affiliations:** 1Department of Nutrition and Food Sciences, School of Arts and Sciences, Lebanese International University, Beirut P.O. Box 146404, Lebanon; suzan.haidar@liu.edu.lb; 2Faculty of Health, Medicine and Life Sciences, Maastricht University, P.O. Box 6166200, 6211 LK Maastricht, The Netherlands; n.devries@maastrichtuniversity.nl; 3Laboratory of Dietetics and Quality of Life, Department of Food Science and Human Nutrition, Agricultural University of Athens, 11855 Athens, Greece; lpoulia@gmail.com; 4Nutrition Program, Department of Natural Sciences, Lebanese American University, Beirut P.O. Box 135053, Lebanon; hussein.hassan@lau.edu.lb; 5Department of Biomedical Sciences, School of Arts and Sciences, Lebanese International University, Beirut P.O. Box 146404, Lebanon; mohammad.rached@liu.edu.lb; 6School of Nutrition, Faculty of Community Services, Toronto Metropolitan University, Toronto, ON M1L2A6, Canada

**Keywords:** neck circumference, anthropometry, metabolic syndrome

## Abstract

Metabolic syndrome (MetS) is a cluster of symptoms that, when present, increase the risk for cardiovascular disease. There is a need for reliable screening tools that are ethnically sensitive. Two hundred and sixty-six college students were assessed anthropometrically. They had a fasting blood sample drawn, and blood pressure measured. They then completed a demographic questionnaire and The International Physical Activity Questionnaire (IPAQ). The prevalence of MetS was found to be 10.1% in males and 4.5% in females. Neck circumference (NC) was positively associated with BMI in males (r = 0.55, *p* < 0.001) and females (r = 0.53, *p* < 0.001) and was positively associated with hip circumference in both males (r = 0.47, *p* < 0.001) and females (r = 0.50, *p* < 0.001) and with waist circumference in males (r = 0.46, *p* < 0.001) and females (r = 0.49, *p* < 0.001.) An area under the curve (AUC) was calculated using receiver operating characteristics (ROC), and NC > 38 cm in males and NC> 36 cm in females were found to be appropriate cut-offs for diagnosing MetS. NC is a reliable and non-invasive screening tool that can be used to screen for MetS in males. NC could also serve as an anthropometric instrument to assess abdominal obesity and could be valuable for college students.

## 1. Introduction

Obesity is defined as having an abnormal surplus of body fat, thus putting those inflicted by it at increased risk for morbidity and mortality. Globally, obesity rates have almost tripled since the year 1975 and have reached pandemic proportions despite intensive public health interventions for both prevention and management [1].

The main burden of obesity lies in the comorbidities associated with it [2,3,4], notably metabolic syndrome (MetS), which is a condition characterized by elevated waist circumference (WC) (visceral fat accumulation), dyslipidemia, insulin resistance and increased blood pressure(BP) [5,6]. Patients with MetS are at greater risk of both cardiovascular disease (CVD) and diabetes [5,7].

In order to evaluate body composition and identify, prevent, control and/or treat obesity and its comorbidities, such as MetS, reliable and accurate anthropometrics may be used. Body mass index (BMI), the most commonly used tool for the identification of overweight and obesity, has major drawbacks, such as its inability to identify the location of adipose tissue accumulation or the body composition, as increased body mass does not always correspond to increased fat mass [8,9,10]. For instance, an abnormal BMI can be found in athletes with increased muscle mass, such as bodybuilders, and a normal BMI can mask sarcopenia in older people [11].

In order to increase the efficiency and the sensitivity of anthropometric markers in routine measurements to detect visceral fat, several measurements have been studied. Neck circumference (NC) has received recent attention due to its practicality and ease of measurement. The NC has been proven to be able to identify excess upper body adipose tissue and seems to correlate well with increased risk of CVD [12]. It is an easy and inexpensive technique and, unlike waist circumference, does not lead to diurnal variation [13]. Moreover, it can bypass the cultural barriers of removing clothes of the upper body for more accurate measurement. This allows us to consider NC a viable marker for android adiposity risk assessment and thus MetS for Arab countries, and communities where veils are worn [12].

Therefore, the aim of our study was to identify whether NC can be used as an alternative to WC in identifying patients with central fat distribution and increased risk of MetS and to determine the appropriate ethnic-specific cut-offs for the Lebanese adult population.

## 2. Methods

### 2.1. Sampling

The participants in this study were recruited from four selected university campuses in different geographical locations in Lebanon. Campuses were specifically chosen to cover both rural and urban areas in Lebanon. Initially emails were sent to all students (approximately 20,000 students), and classroom visits were carried out by researchers to explain the study protocol and its objectives to potential participants. The participants were informed about their right to withdraw from the study at any time. Those who were interested in pursuing the study were later contacted and given appointments at the nutrition clinic. Inclusion criteria for participation in the study were that participants had to be enrolled in a university program and were adults. Students were excluded from the study if they were sick at the time of the study, suffered from chronic diseases, or were not fasting at the time of testing. Students who did not meet inclusion criteria were excluded from participation. A total of 266 consenting students met the inclusion criteria and agreed to take part in the study. Ethical approval for this study was obtained from the Lebanese International University (case number: LIUCRE-141117-2).

Data collection took place at the nutrition clinics on the four different campuses. Students were contacted the day before data collection and reminded about the necessity of coming in after a 12 h fast.

### 2.2. Minimal Sample Size Calculation

A minimum of 29 students was deemed necessary to have adequate statistical power, based on a 5% risk of error, 95% power and using G-power software.

### 2.3. Anthropometrics

Trained licensed dietitians collected the students’ anthropometric data while standing in the Frankfort plane position. Height (cm) was measured while participants were barefoot and was rounded to the nearest 0.1 cm, using a portable stadiometer (ADE stadiometer, Hamburg, Germany). Weight was measured with minimum clothing and without shoes on a calibrated beam scale (Detecto, Webb City, MO, USA.) and rounded to the nearest 100 g.

Waist circumference (cm) was measured at the mid-point, half-way between the right iliac crest and the lower costal region [14], and neck circumference (cm) was measured at the middle of the neck, between the mid-cervical spine and mid anterior neck [15]. Hip circumference was also measured at the most prominent point between the waist and the thigh [16]. Waist, hip and neck circumferences were measured to the nearest 0.1 cm, using Accugirth calibrated measuring tapes. WC was considered elevated when equal to or above 94 cm in males and equal to or above 80 cm in females [6].

ΒMI was calculated using the equation Body Weight/ Height^2^ (kg/m^2^), and BMI categories were defined as <18.5, 18.5–24.99, 25–29.9 and >30 kg/m^2^ for underweight, normal, overweight and obese, respectively [16].

### 2.4. Biochemical Markers

A 5 mL blood sample was drawn by a licensed phlebotomist and analyzed at an external laboratory. Blood was collected between 7:00 am and 10:00 am. Serum was analyzed for lipids (total cholesterol (mg/dL)), high-density lipoprotein (HDL-cholesterol) (HDL-C; mg/dL), triglyceride (TG; mg/dL) and low-density lipoprotein (LDL-cholesterol (LDL-C; mg/dL.) The blood was also analyzed for high-sensitivity C-reactive protein (CRP; mg/L), cortisol (nmol/L) and fasting blood glucose (FBG; mg/dL) concentrations. The analyses of total cholesterol, HDL-C, TG, CRP and FBG concentrations were performed using the Cobas C111 automated biochemical analyzer (Roche Diagnostics, Indianapolis, IN, USA) based on spectrophotometric principles. LDL-C levels were calculated by the Friedewald equation if triglyceride levels were below 400 mg/dL. Serum cortisol (nmol/L) morning level was measured using Cobas e411 immunoassay automated analyzer (Roche Diagnostics, Indianapolis, IN, USA) based on electrochemiluminescence (ECLIA) principle.

Impaired FBG was defined as blood glucose between 100 and 125 mg/dL, HDL was considered low when the value was less than 40 mg/dL in males and less than 50 mg/dL in females (Roche kit, USA). LDL value was considered high when the value was equal to or above 130 mg/dL and the classification of high total cholesterol was established when the value was equal to or above 200 mg/dL (Roche kit, Little Falls, NJ, USA.) Elevated TG was classified as such when the value was equal to or above 150 mg/dL. CRP was considered elevated above 5 mg/dL (Roche kit, Little Falls, NJ, USA), and cortisol levels were considered high when the value was above 536 nmol/L (Roche kit, Little Falls, NJ, USA).

### 2.5. Blood Pressure Measurement

Blood pressure was measured in the seated position after 5 min of rest by trained research assistants, using a standardized mercury sphygmomanometer [17]. The average of two repeated readings of systolic blood pressure (SBP) and diastolic blood pressure (DBP) were taken on the same arm within a two-minute time interval [17]. Elevated SBP was defined as BP equal to or above 120 mmHg, whereas elevated DBP was considered as BP equal to or above 80 mmHg [17].

### 2.6. Metabolic Syndome

MetS was diagnosed based on the International Diabetes Federation (IDF) guidelines, which specify that MetS exists in patients with abdominal obesity where waist circumference is ≥80 cm in females and ≥94 cm in males in addition to two of the following criteria: (a) SBP ≥ 130 mmHg or DBP ≥ 85 mmHg (or treatment with an antihypertensive medication), (b) triglycerides levels (TG) ≥ 150 mg/dL or on treatment; (c) HDL-cholesterol levels (HDL-C) <40 mg/dL in males and <50 mg/dL in females, (d) raised fasting blood glucose ≥ 100 mg/dL [6].

### 2.7. Questionnaires

The students were then asked to fill out the following questionnaires with the help of research assistants:(1)**Demographic and lifestyle habits questionnaire:** adapted from Levitsky et al. 2004 [18], had 10 open-ended questions assessing number of meals consumed per day and frequency of eating outside the home, living arrangements, smoking status and alcohol consumption.(2)**The International Physical Activity Questionnaire (IPAQ)** 2014 [19] short form: a validated tool used to assess level of physical activity, which consists of seven questions designed to measure both duration and frequency exercise of light, moderate and vigorous physical activity completed by participant in the past week. Metabolic equivalent of tasks (MET) was then calculated by multiplying the total minutes spent participating in the activity by the frequency (days) and the constants of 3.3, 4.0 and 8.0 for light, moderate and vigorous activity, respectively. The total MET value was then computed by summing up the respective MET values for all activities that were carried out in bouts longer than 10 min in duration.

All questionnaires were available in both the English and Arabic languages and were pilot-tested on a sample of 20 students before the study that was conducted for validation and eradicating ambiguity. The questionnaires used in piloting were not included in this study. The total duration of data collection from the students took between 30 and 50 min.

## 3. Statistics

SPSS version 21 was used for data analysis. Descriptive statistics was conducted for all measured variables. Categorical variables were expressed as frequencies and percentages, whereas continuous variables were expressed as mean ± standard deviation. Normality was assessed using the Kolmogorov–Smirnov test. Bivariate analysis between continuous variables was assessed using Mann–Whitney U test, whereas the chi-square test was used for bivariate analysis of categorical variables. The Spearman correlation coefficient was used to evaluate the relationship between NC and metabolic variables. NC cutoff values for determining the MetS was determined using ROC curve analysis using MedCalc software (De Long method). The results of the analysis used the Youden index are reported as sensitivity, specificity, positive predictive value, negative predictive value, likelihood ratios and area under the curve (AUC). *p* values lower than 0.05 were considered statistically significant.

## 4. Results

A total of 266 participants were recruited for this study, 89 of whom were male (33.5%) and 177 female (66.5%), Among the participants, 48 males (18%) and 100 females (38%) were found to have a healthy BMI, whereas 24 males (9%) and 44 females (16%) were found to be overweight. A total of 97 participants (36.8%) were found to be either overweight or suffering from obesity. The baseline characteristics for the study sample are shown in Table 1.

MetS was prevalent in 10.1% of males and 4.5% of females. The anthropometric markers BMI, WC and NC were significantly higher in males than females. Similarly, males had significantly higher levels of LDL-cholesterol, triglycerides, blood glucose CRP, cortisol, SBP, DBP and physical activity, whereas females had higher levels of HDL-cholesterol. There were no significant differences between males and females for hip circumference, total cholesterol or CRP (Table 1).

All participants who suffered from MetS had a significantly higher NC than those who did not (38.56 ± 3.90 vs. 35.92 ± 3.29, respectively, *p* = 0.006) (Table 2).

In order to determine sex-specific cut-off values for NC to predict MetS, ROC curve analysis was performed and is presented in Figure 1. The identified NC for the diagnosis of MetS for males was found to be >38 cm (AUC: 0.738, 95% confidence interval = 0.634–0.826, *p* = 0.016) with a sensitivity of 88.89% and specificity of 53.75 %.

The cut-off NC for females was found to be >36 cm (AUC: 0.611, 95% confidence interval = 0.535–0.683, *p* = 0.3378) with a sensitivity of 50 % and specificity of 73.96% (Table 3 and Table 4).

When NC was correlated with the anthropometric indices while controlling for gender, we found out that NC was positively associated with BMI in males (r = 0.55, *p* < 0.001) and females (r = 0.53, *p* < 0.001). It was also positively associated with hip circumference (r = 0.47, *p* < 0.001) in males and females (r = 0.50, *p* < 0.001) and with WC in males (r = 0.46, *p* < 0.001) and females (r = 0.49, *p* < 0.001) (Table 5).

## 5. Discussion

In this cross-sectional study, we were able to show that NC is a potential tool that can be used to screen for obesity and MetS. NC was significantly greater in males suffering from MetS versus those who did not have MetS (38.56 ± 3.90 vs. 35.92 ± 3.29, respectively). However, such a relationship was not found to be as significant in females, and sensitivity for NC as a marker for MetS may be poor for females. We also found that NC was positively correlated with all the other anthropometric markers currently used for obesity screening, which include WC, hip circumference and BMI. Moreover, we were able to determine that an NC greater than 38 cm in males and higher than 36 cm in females suggests the presence of the MetS.

Several studies have been published with the aim of identifying appropriate cut-offs for NC to screen for MetS. In a study conducted in Thailand, NC ≥ 38 cm in males and NC ≥ 33 cm were found to be a valid predictor for MetS [20]. Moreover, an NC of 38 cm for men and 37 cm for women was the best cut-off point for determining MetS in a Turkish population [21]. Our cut-off points are similar to those of both the Turkish and Thai studies.

Furthermore, among Brazilian participants, an NC > 40 cm and > 36.1 cm for males and females, respectively, predicted MetS [22] and among Chinese participants, NC ≥ 37 cm for men and NC ≥ 33 cm for women were the best cut-off points for MetS [23]. Since the numbers are similar across different studies, it is safe to assume that our results are true and applicable in the Lebanese. Of course, slight disparity is normal and expected, as is the case for WC among different ethnicities [24]. It is also possible that the slight variation could be due to the procedure used to measure NC, as currently there is no consensus about the exact location of the measurement position [25,26,27,28].

In our study, we found that NC was only significantly correlated with LDL in both genders. NC was only significantly associated with triglycerides in males and total cholesterol in females, Furthermore, we found that SBP was only significantly correlated with NC in females. This is unlike other studies that did actually find correlation between blood lipids, blood pressure and blood glucose markers and NC. For example, Preis et al., (2010) found that NC correlated with all cardiometabolic risk factors except for those of total and LDL cholesterol [25]. Additionally, Laohabut and colleagues (2019) found that there was a positive correlation between NC, BP and LDL-C in both sexes [20]. Moreover, an analysis carried out on the Framingham Heart Study data established that the NC was associated with elevated blood pressure, dyslipidemia and insulin resistance, all of which are markers for MetS [29]. Our results are not in line with these findings, and we hypothesize that this could be due to the fact that our participants were college students, who tend to be younger and healthier than older adults.

Furthermore, in our sample, NC was a better predictor of MetS in males (AUC = 0.738) than in females (AUC = 0.611). This could be because, on average, the females in our sample had a normal BMI, whereas the males were overweight.

We are aware that our study has several limitations; first, our sample consisted of students and is not representative of the entire Lebanese population. Additionally, NC is a proxy measure for upper body fat, and we did not actually measure obesity by any quantifiable method, as we used only BMI and WC. However, we found a correlation between NC and WC, which is validated for abdominal obesity screening [30]. Additionally, this study was a cross-sectional study, and limitations exist with such a study design. Additionally, the percentage of females in the sample was higher than that of males; however, this was the case with this specific university’s enrollment. We also accept that there is a possibility of self-selection bias among the participants.

Despite the limitations, our study had several strengths; first, research assistants were extensively trained prior to the beginning of the study to systematically obtain the anthropometric data. Second, this is the first study conducted in Lebanon and, to our knowledge, also the first study to verify the effectivity of NC as a screening tool for MetS screening.

An increasing number of studies have suggested that NC can be used as a simple and inexpensive anthropometric marker for obesity and MetS [31,32,33]. NC could be superior to WC as a marker for visceral fat [34,35]. Some researchers have criticized WC’s validity and reproducibility because of intraindividual variation, which has been reported to fall anywhere between 0.3 and 4.7 cm, sometimes reaching 8 cm. These fluctuations could result from respiration-related variations, food ingestion and depression of the abdomen, hence potentially giving false-negative results [36]. One other advantage of the NC is that patients are not required to take off any of their clothes, which may be a problem for those who wear veils or have body image issues.

## 6. Conclusions

In conclusion, NC could be a promising anthropometric marker for visceral fat and abdominal obesity. It could also be used as a less invasive tool to screen for MetS in both community and clinical settings. However, for a robust association, future large-scale and representative studies should be carried out to determine cut-offs among all age groups and sexes.

## Figures and Tables

**Figure 1 diseases-10-00031-f001:**
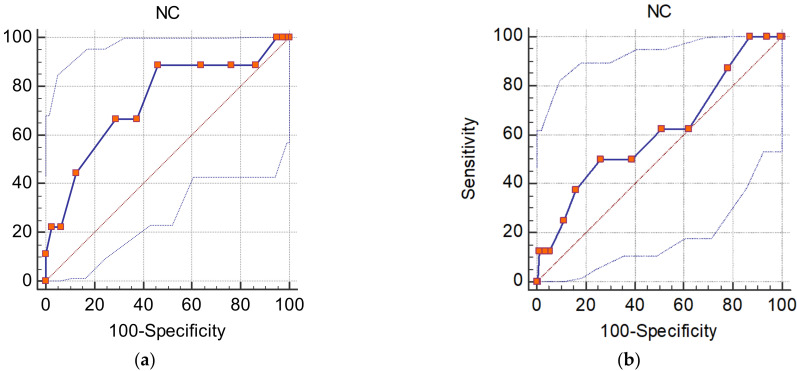
Receiver operating characteristic (ROC) curves show the specificity and sensitivity of neck circumference for predicting metabolic syndrome in males (**a**) and females (**b**).

**Table 1 diseases-10-00031-t001:** Baseline demographic, lifestyle, anthropometric, clinical and biochemical characteristics of participants (*n* = 266).

	Male (89; 33.5%)	Female (177; 66.5%)	
**Parameter**	**Mean ± SD**	**Mean ± SD**	***p*-Value**
Age (years)	20.61 ± 3.15	20.54 ± 3.03	0.899
BMI (kg/m^2^)	25.01 ± 4.48	23.68 ± 4.56	0.016
Hip circumference (cm)	101.80 ± 11.83	100.82 ± 10.31	0.776
Neck circumference (cm)	38.68 ± 2.75	34.79 ± 2.90	<0.001
Waist circumference (cm)	89.51 ± 11.34	83.74 ± 11.04	<0.001
LDL-cholesterol (mg/dL)	100.17 ± 32.70	91.85 ± 27.26	0.049
Total cholesterol (mg/dL)	162.87 ± 35.62	162.97 ± 28.14	0.726
Triglycerides (mg/dL)	96.70 ± 53.48	74.20 ± 31.10	<0.001
HDL-cholesterol (mg/dL)	45.18 ± 9.27	57.08 ± 12.90	<0.001
Blood glucose (mg/dL)	86.61 ± 20.01	80.59 ± 6.79	<0.001
C-reactive protein (mg/)	2.44 ± 3.21	1.83 ± 2.45	0.019
Cortisol (nmol/L)	494.51 ± 158.19	387.86 ± 188.47	<0.001
Systolic blood pressure (mmHg)	120.45 ± 10.33	100.79 ± 10.33	<0.001
Diastolic blood pressure (mmHg)	7.57 ± 1.21	6.84 ± 0.90	<0.001
Physical activity (METs)	2777.93 ± 3255.35	1164.05 ± 1611.69	<0.001
	***n* (%)**	***n* (%)**	
Smoking status (Yes)	37 (41.6%)	28 (15.8%)	<0.001
Alcohol consumption (Yes)	17 (19.1%)	12 (6.8%)	0.002
Metabolic syndrome (Yes)	9 (10.1%)	8 (4.5%)	0.078

**Table 2 diseases-10-00031-t002:** Neck circumference compared between those with and without metabolic syndrome (*n* = 266).

	Metabolic Syndrome		
	Absence (*n* = 249)	Presence (*n* = 17)	*p*-Value
**Male NC (cm)**	38.45 ± 2.65	40.78 ± 2.89	0.017
**Female NC (cm)**	34.73 ± 2.87	36.06 ± 3.42	0.298
**All participants NC (cm)**	35.92 ± 3.29	38.56 ± 3.90	0.006
**BMI**	23.79 ± 4.30	29.11 ± 5.64	<0.001

NC = neck circumference; BMI: body mass index.

**Table 3 diseases-10-00031-t003:** Performance of neck circumference for predicting MetS in males.

NC (cm)	Sensitivity	Specificity	PPV	NPV	LR+	LR−	AUC
≥31	100	0	10.1	-	1.00	-	0.899
>31	100	1.25	10.2	100	1.01	0	0.888
>33	100	2.50	10.3	100.0	1.03	1.03	0.876
>34	100	5.00	10.6	100.0	1.05	0.00	0.854
>35	88.89	13.75	10.4	91.7	1.03	0.81	0.787
>36	88.89	23.75	11.6	95.0	1.17	0.47	0.697
>37	88.89	36.25	13.6	96.7	1.39	0.31	0.584
>38	88.89	53.75	17.8	97.7	1.92	0.21	0.427
>39	66.67	62.50	16.7	94.3	1.78	0.53	0.371
>40	66.67	62.50	16.7	94.3	1.78	0.53	0.371

PPV: positive predictive value; NPV: negative predictive; LR+: positive likelihood ratio; LR−: negative value; AUC: area under curve.

**Table 4 diseases-10-00031-t004:** Performance of neck circumference for predicting MetS in females.

NC (cm)	Sensitivity	Specificity	PPV	NPV	LR+	LR−	AUC
≥29	100	0	4.5	-	1.00	-	0.955
>29	100	0.59	4.5	100	1.01	0.00	0.949
>30	100	5.92	4.8	100.0	1.03	0.00	0.898
>31	100	13.02	5.2	100.0	1.15	0.00	0.831
>32	87.50	21.89	10.4	91.7	1.12	0.57	0.751
>33	62.50	37.87	4.5	95.5	1.01	0.99	0.610
>34	62.50	49.11	5.5	96.5	1.23	0.76	0.503
>35	50.00	60.95	5.7	96.3	1.28	0.82	0.395
>36	50.00	73.96	8.3	96.9	1.92	0.68	0.271
>37	37.50	84.02	10.0	96.6	2.35	0.74	0.181

PPV: positive predictive value; NPV: negative predictive; LR+: positive likelihood ratio; LR−: negative value; AUC: area under curve.

**Table 5 diseases-10-00031-t005:** Correlation analysis of neck circumference and anthropologic indices and metabolic risk factors compared between males and females (*n* = 266).

	Neck Circumference					
	Males		Females		All	
	r	*p*-Value	r	*p*-Value	r	*p*-Value
Age (years)	0.04	0.697	−0.006	0.932	0.01	0.863
BMI (kg/m^2^)	0.55	<0.001	0.53	<0.001	0.52	<0.001
Hip circumference (cm)	0.47	<0.001	0.50	<0.001	0.41	<0.001
LDL-cholesterol (mg/dL)	0.18	0.082	0.18	0.012	0.22	<0.001
Total cholesterol (mg/dL)	0.23	0.024	0.13	0.64	0.13	0.34
C-reactive protein (mg/L)	0.18	0.083	0.20	0.007	0.25	<0.001
Cortisol (nmol/L)	−0.02	0.845	−0.06	0.411	0.14	0.018
Physical activity (METs)	−0.04	0.677	−0.007	0.922	0.13	0.027
Waist circumference (cm)	0.46	<0.001	0.49	<0.001	0.51	<0.001
Systolic blood pressure (mmHg)	0.10	0.343	0.18	0.012	0.40	<0.001
Diastolic blood pressure (mmHg)	0.15	0.143	0.08	0.260	0.26	<0.001
Blood glucose (mg/dL)	−0.05	0.622	0.23	0.002	0.26	<0.001
Triglycerides (mg/dL)	0.31	0.003	0.06	0.380	0.24	<0.001
HDL-cholesterol (mg/dL)	−0.05	0.59	−0.22	0.003	−0.39	<0.001

## Data Availability

Data available from authors upon request.

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
