# Peer review of "Neck Circumference as a Screening Tool for Metabolic Syndrome among Lebanese College Students"

_diseases, 2022, doi:10.3390/diseases10020031_

Round 1

Reviewer 1 Report

Introduction.

  1. The objectives of the study do not indicate the role of NC to identify risk of metabolic syndrome. However, this relationship is one of the contributions of the work.

Methods.

  1. Line 63. “Emails were sent to all students” ¿How many students received the email?
  2. ¿Did you calculate the sample size?
  3. Biochemical markers. The criteria for the diagnosis of metabolic syndrome is not included.

Results.

  1. Table 1.

Please correct the concept “Waist Neck Circumference”.

Blood Pressure (systolic and diastolic) score is wrong.

“Stress Score” and “Sleep Quality” scores appear but it is not indicated how they were obtained. Nor is it stated (Methods) what was the criteria for making the diagnosis of Metabolic Syndrome.

Based on the BMI, please indicate how many subjects were overweight or obese.

  1. The paragraph between lines 167 and 176 repeats all the results that are already in Table 1. It could be reduced or deleted.
  2. Table 2. I suggest completing the table with the average BMI values and evaluating the differences between subjects with or without MetS; the Authors could observe if the behavior is similar to the NC. BMI has traditionally been used to identify increased risk of MetS.
  3. Tables 3 and 4. It is necessary to note in the foot of the table the meaning of the abbreviations PPV, NPV, LR+ and LR-
  4. Table 5. Please note that the correlation test is Spearman's (according to the statistics section).

Discussion

  1. Line 220. “NC was significantly greater in patients with MetS versus those without Mets” Please specify that it was presented only in men.
  2. Line 224-225. Please clarify and discuss that the sensitivity is poor, particularly in women, which could result in the non-identification of subjects with MetS; contrary to men in whom the specificity is low. And also note it in the Conclusions.
  3. Line 239-240. The authors wrote “In our study we found that NC was not significantly correlated with any biochemical marker”; but Total Cholesterol and Triglycerides had a p value of 0.02 and 0.003 respectively (males). LDL-C and C-Reactive protein were almost significant (p= 0.08) also in males.
  4. In women, LDL (p= 0.01) C-Reactive protein (p= 0.007), Systolic BP (p= 0.01), Blood glucose (p= 0.002) and HDL-C (p= 0.003) values showed significant correlations.

Conclusions.

  1. With the results shown here, it is not possible to conclude that “NC could be a reliable and superior anthropometric marker for visceral fat and abdominal obesity”.
  2. And with respect to its use for MetS screening, the differences between genders should be mentioned.

Reviewer 2 Report

This study  identifies whether NC can be used as an alternative of waist circumference in identifying patients with central distribution of fat, android obesity and cardiometabolic risk. College students were assessed anthropometrically and the prevalence of Metabolic Syndrome was found to be 10.1% 16 of males and 4.5% of females.  Neck circumference (NC) is a reliable and non-invasive screening tool that can be used 22 to screen for Metabolic Syndrome.

COMMENTS

It is appropriate to explain the NC abbreviation in abstract.

Methods - 2.2 Anthropometrics - line 86: reference n.16 appears before reference 14

Methods - 2.4 - “Elevated BP was defined as blood pressure equal to or above 120/80 mmHg”: it is appropriate to indicate which reference this cutoff suggests.

Results

How many people are normal and overweight in both sexes?

Table 1: the fifth parameter is Waist Neck Circumference?

Table 1: Systolic and Diastolic BP: Are the values correct?

Table 1: stress score and sleep quality: how are they evaluated?

Table 2: It is appropriate to enter the number of subjects

Table 5: HDL-Cholesterol for Females: is the “r” value correct?

Round 2

Reviewer 1 Report

The suggestions for corrections that I made to the original manuscript have been carried out. 

Author Response

Many thanks to the reviewers for their comments.

This manuscript is a resubmission of an earlier submission. The following is a list of the peer review reports and author responses from that submission.